# Exploring Heading Direction Perception in Cervical Dystonia, Tremor, and Their Coexistence

**DOI:** 10.3390/brainsci14030217

**Published:** 2024-02-27

**Authors:** Aratrik Guha, Hanieh Agharazi, Palak Gupta, Aasef G. Shaikh

**Affiliations:** 1Department of Biomedical Engineering, Case Western Reserve University, Cleveland, OH 44106, USA; axg1334@case.edu (A.G.); pxg239@case.edu (P.G.); 2National VA Parkinson Consortium Center, Neurology Service, Louis Stokes Cleveland VA Medical Center, Cleveland, OH 44106, USA; hxa134@case.edu; 3Department of Neurology, Case Western Reserve University, Cleveland, OH 44106, USA; 4Movement Disorders Center, Neurological Institute, University Hospitals, Cleveland, OH 44106, USA

**Keywords:** dystonia, tremor, dystonic tremor, tremor associated with dystonia, basal ganglia, cerebellum, balance, postural abnormality

## Abstract

Objective: Dystonias, characterized by excessive muscle contractions resulting in involuntary postures and movements, impact 3 million people globally, making them the third most common movement disorder. Often accompanied by tremors, dystonias have epidemiological links and non-motor features shared with isolated tremor, such as essential tremor. Both dystonia and tremor present with balance dysfunction and abnormal involuntary movements, potentially linked to abnormal cerebellar function. This study explores the perception of one’s own linear movement, heading, particularly discrimination of heading direction, in isolated cervical dystonia, isolated tremor, and their combination. We compare such perception behavior in visual and vestibular domains, predicting that visual heading perception would be superior to vestibular heading perception. Methods: Following the focus on the perception of heading direction, we used psychophysics techniques, such as two-alternative-forced-choice task, to examine perception of direction of one’s own movements as they see isolated visual star-cloud movement (visual heading perception) and en bloc body movement (vestibular heading perception). We fitted a sigmoidal psychometric function curve to determine the threshold for visual or vestibular heading perception in our participants. Results: Nineteen participants underwent a two-alternative forced-choice task in the vestibular and visual domains. Results reveal elevated vestibular heading perception thresholds in cervical dystonia with or without tremor, and isolated tremor compared to healthy controls. Vestibular heading perception threshold was comparable in cervical dystonia with tremor and isolated tremor, but it was even worse in isolated cervical dystonia. Visual heading perception, however, remained less affected all three conditions—isolated cervical dystonia, isolated tremor, and their combination. Conclusion: These findings indicate shared deficits and distinctions in the perception of linear translational heading across movement disorders, such as isolated cervical dystonia, tremor, or their combination, offering insights into their pathophysiology, particularly the involvement of cerebellum regions responsible for vestibular processing.

## 1. Introduction

The dystonias are disorders characterized by excessive muscle contractions leading to involuntary postures and movements with a repetitive quality, resulting in jerky oscillations [1,2]. Dystonia affects 3 million people worldwide and is third most common movement disorder following tremor and Parkinson’s disease. Tremors are defined by regular oscillations of a body region, typically with a sinusoidal pattern [3,4]. Tremor is the most common of all movement disorders, affecting >20 million people worldwide affecting approximately 0.3% prevalence of the general population. Tremor is a progressive disease leading to more disability at older age. As with the dystonias, tremors are chronic, and patients face a lifetime of overt stigmatizing disability. Although dystonia and tremor are considered distinct disorders, they are closely related. Epidemiologically, many patients diagnosed with dystonia also have tremor. The reported prevalence of clinically apparent tremor among patients diagnosed with dystonia varies from 10–90%, with an overall weighted average of 35.1% [5]. Conversely, the patients diagnosed with essential tremor also have dystonia, with reported co-prevalence rates of up to 21% [5]. 

Dystonia and tremor, although traditionally considered motor system disorders, also share non-motor features. The non-motor features in essential tremor include deficits in executive function, attention, concentration, and memory. The essential tremor patients also present with impairments in special senses, in the form of impaired color vision, olfaction, and hearing abnormalities [6,7,8]. Sleep dysregulation and social anxiety is also reported in essential tremor [9]. The neuropsychological deficit in essential tremor is thought to be related to cerebellar cognitive affective [10]. Among the non-motor deficits observed in cervical dystonia (CD) patients is the inability to mentally manipulate their surrounding space [11]. Furthermore, CD patients experience difficulties in complex movement planning, motor dexterity, visual–spatial working memory, and tactile object recognition [11,12].

Balance dysfunction is not uncommon in essential tremor and dystonia, and is often attributed to abnormal cerebellar function. Cervical vestibular-evoked myogenic potentials and ocular vestibular-evoked myogenic potentials—examining the vestibulo-collic and vestibulo-ocular reflex have increased amplitude but shortened latency in essential tremor compared to healthy controls [13]. Abnormal vestibulo-collic reflex in essential tremor suggests putative dysfunction in regions that process central vestibular information, such as the cerebellum [14]. Balance dysfunction is commonly seen in cerebellar predominant form of essential tremor where action tremor is much more robust, and it is often associated with dysarthria in this subset of patients [15]. Cerebellar predominant essential tremor also manifests with dysfunction of higher-order processing of vestibulo-ocular reflex that localize to caudal vermis [16]. The visuospatial skills are notably abnormal in CD, as evidenced by impaired performance in tasks such as line bisection, spatial quadrant selection, subjective perception of verticality, and drawing [17,18]. Additionally, approximately one-third of CD patients lack awareness of their own neck posture and exhibit proprioceptive dysfunction, along with an increased threshold of tactile temporal discrimination [19]. Despite the prevalence of these visuospatial dysfunctions in CD, the underlying biological mechanisms remain largely unknown [20]. 

Heading perception is a non-motor skill crucial for functional spatial navigation and encompasses the ability to perceive one’s own motion in relation to their immediate environment while relying on sensations from the vision and vestibular systems. Our recent investigation of spatial navigation performance involving linear-translational heading in CD challenged the idea that CD-related vestibular dysfunction is due to peripheral vestibular deficits. Our study showed abnormal proprioceptive input from neck muscle inducing the noise into the central mechanism, integrating visual, vestibular, and proprioceptive signals. We found that heading perception thresholds are elevated in CD, and the worsening of threshold correlates with the severity of CD. The fundamental question remains as to whether impaired vestibular and visual heading perception in CD is dependent upon presence (or absence) of associated tremor, and that whether such performance is different if the participant does not have any other movement disorder but isolated tremor.

Here we examine the degree of heading perception performance for vestibular and visual heading, while interpreting threshold and biases for isolated CD without tremor (or any other movement disorder), CD with tremor, and isolated tremor without any dystonia (or other movement disorder). The hypothesis is that the visual heading perception is better than vestibular perception in isolated CD, isolated tremor, and CD with tremor. The vestibular heading perception is comparable in isolated tremor and CD with head tremor, while both conditions would differ from isolated CD. 

## 2. Materials and Methods

### 2.1. Participants

Observations in the performance of visual and vestibular discrimination were made for 19 participants (6 with isolated CD, 3 with CD and tremor affecting the neck and/or arms; 5 with isolated tremor of the arms, and 5 healthy controls). The tremor was defined as to-and-fro regular rhythmic oscillations that approximates to a sinusoidal waveform [21]. The oscillatory movements with jerky, coarse, irregular features, combined with drifts and corrective movements were not considered as tremor; when present they were considered dystonia [22]. The tremor and dystonia patients considered in this study did not have any other movement disorders. We also excluded those with dementia, clinically significant depression or anxiety, focal brain lesions, intracranial surgery (except deep brain stimulation) history, atypical parkinsonism or Parkinson’s disease, and generalized, multifocal, or tardive dystonia were excluded. None of our patients had clinical evidence for functional movement disorder. All CD participants were successfully treated with botulinum toxin, while one participant was implanted with a pallidal deep brain stimulator. At baseline the patients were adequately treated with pharmacotherapy or deep brain stimulation. If the patients were receiving botulinum toxin, the responses were measured on the day when botulinum toxin were injected. The stimulator was turned off prior to measuring the performance in those who had DBS. Healthy controls did not have any known neurological or visual disorders (Table 1). Participants were recruited from the National VA Parkinson’s Consortium Center at Cleveland VA Medical Center. 

### 2.2. Experimental Setup

All participants provided informed written consent per the Declaration of Helsinki and the Institutional Review Boards of the Louis Stokes Cleveland VA Medical Center.

Vestibular heading discrimination was assessed using a two-alternative forced-choice task with Meta’s Oculus controllers (Oculus Rift, Menlo Park, CA, USA). Participants were secured in a six-degree-of-freedom motion platform (hexapod) from Moog (Aurora, NY, USA), which enabled movement in three-dimensional space. Seated in the hexapod harness, participants had cushioned helmets immobilize their heads (Figure 1A). The experiment took place in a dark room to eliminate external cues. Across three blocks, there were 99 trials with randomized forward motion angles at 0°, 5°, 10°, 20°, and 30° to the left or right (Figure 1A). The hexapod displaced participants 0.4 m over 1.5 s, surpassing vestibular thresholds, then paused for 3 s. During this pause, participants were asked to utilize the handheld controllers to indicate direction of movement.

In the visual heading discrimination task, the participants observed a 3D optical-flow pattern through Oculus Rift VR headsets (Menlo Park, CA, USA) (Figure 1B). There were 169 trials per 12 min block. The trials featured star clouds expanding radially left or right at angles of 0°, 5°, 10°, 15°, 20°, 25°, or 30° (Figure 1B). To respond, participants used a handheld button.

### 2.3. Data Analysis

We analyzed the button clicks during the vestibular and visual heading tasks that participants used to report perceived motion direction. This allowed us to quantify the perception threshold and direction perception bias for each participant. We determined these parameters by fitting a Gaussian cumulative distribution psychometric function to the percentage of rightward decisions at different motion direction levels. The motion direction levels were derived from the binary left/right responses. We defined the vestibular and visual thresholds (σ) as the standard deviation of the Gaussian fit, where lower threshold values indicated heightened sensitivity in detecting subtle heading changes. We gauged direction perception biases (μ) by linearly translating the psychometric function along the *x*-axis, denoting the heading angle equally likely to elicit left or right responses. This reflected asymmetry in perceiving straight motion. 

## 3. Results

The overarching goal of this study was to examine the hypothesis that the visual heading perception is better than vestibular perception in isolated CD, isolated tremor, and CD with tremor. The vestibular heading perception is comparable in isolated tremor and CD with head tremor, while both conditions would differ from isolated CD. This study directly examined the hypothesis and compared a total of 19 human participants: six were isolated CD, three had CD with head tremor, five had isolated tremor, and five matched healthy controls. A two-alternative forced-choice task assessed perceived movement direction during passive forward motion (vestibular heading perception) and perceived direction of optic flow across different angular deviations from center in virtual reality (visual heading perception). As anticipated, larger eccentric angles correlated with more accurate direction identification responses, but the accuracy reduced as the direction of heading approached close to straight ahead. Straight-ahead motion was associated with random report of rightward versus leftward motion.

The relationship between the probability of correct responses and eccentricity of heading in visual or vestibular domain yields a sigmoid curve. Such a sigmoid relationship is clearly evident in healthy controls, and it was much robust in the case of visual heading compared to vestibular in healthy participants (gray lines, Figure 2A–F). The visual heading perception was least affected by any of the three conditions (green pink and blue lines in Figure 2A,C,E); however, effects of CD with or without tremor and isolated tremor was much significant on vestibular perception (green pink and blue lines in Figure 2B,D,F). The psychometric function curve depicting the vestibular heading perception almost had an oblique line instead of sigmoid fit in case of the group with CD without tremor (green line Figure 2B). The sigmoid fits of CD with tremor and isolated tremor groups were comparable to each other (pink and blue lines, Figure 2D,F). Nevertheless, in all three disease types the vestibular heading perception psychometric function curve robustly differed from healthy controls. 

Further analysis examined the differences in sigmoid fit parameters, namely threshold and bias, in case of isolated CD, CD with tremor, and isolated tremor in compared them with healthy controls. The value of threshold, the measure of response precision (angle when 75% of correct responses are obtained) is much higher if perception is less precise. The visual heading threshold in case of healthy participants was 4.3° ± 1.9° and it was comparable to isolated CD 5.6° ± 2.1°, CD with tremor 7.6° ± 3.5°, and isolated tremor 5.1° ± 1.6°. The values of vestibular heading threshold in healthy participants (17.6° ± 6.1°) was lower compared to isolated CD (94.4° ± 21.6°). The threshold values in CD with tremor and isolated tremor were 33.5° ± 3.4° and 27.6° ± 7.7° respectively. They were better (smaller) compared to isolated CD, but comparable among CD with tremor and isolated tremor (Figure 3). Given small number of samples, our study could only afford trend-wise comparison, skipping statistical analyses.

The lack of sigmoidal characteristics of vestibular heading perception is associated with unreliable measures of bias. The latter was, however, possible in the case of visual heading perception. The bias in visual heading perception was −2.5° ± 2.5° in patients with isolated CD, 5.6° ± 2.1° in CD with tremor, and −1.5° ± 3.6° in the case of isolated tremor; they were comparable to healthy controls −2.1° ± 4.6°.

## 4. Discussion

Accurate perception of one’s own motion, heading, is critical for maintenance of balance and postural stability [22,23,24,25,26,27]. The perception of verticality in the visual domain and awareness of one’s own posture and walking path when the visual information is compromised are common non-motor impairments in CD. Integration of vestibular, proprioceptive, and visual information via the cerebellum and basal ganglia is critical for maintaining postural stability, balance, and gait [28,29,30]. We recently discovered that CD patients have significant inaccuracy in discrimination of heading direction. The threshold to detect one’s own movement, the heading direction, is higher in CD; the severity is worse in vestibular domain compared to visual [31]. These previous studies, in conjunction with other literature, suggest abnormal vestibular function and vestibulo-ocular reflex, and increased threshold to detect postural vertical in CD [32,33,34,35,36,37,38]. Essential tremor is known to affect balance function. Studies have revealed impairment in vestibulo-ocular reflex, vestibulo-collic reflex, and overall balance impairments in patients with essential tremor [13,14,15,16]. Consistent with previous literature our experiments found impairment in vestibular heading perception in isolated CD, CD with tremor, and isolated tremor.

Why are vestibular and visual heading direction discrimination threshold elevated? 

CD is increasingly thought to be a network-level disorder involving the basal ganglia, cerebellum, brainstem and their connecting pathways [39,40,41]. Likewise, tremor is also a network disorder involving the cerebellum and its connectivity with the thalamus. These regions are extensively connected with the areas responsible for motion perception. Impairment in cerebellar outflow, as predicted in CD or tremor, can influence the cerebellar-thalamic or vestibulo-thalamic projections that are critical for heading direction discrimination, at least in the vestibular [42,43,44]. The vestibular-thalamic projections subsequently relay to the parieto-temporal cortex, playing an important role in vestibular motion perception and visuo-vestibular interaction in visual heading direction discrimination [45,46,47]. Direct alteration of the cerebellar activity or the noise in the activity of the cerebellar projections to the thalamus can therefore adversely impact one’s ability to perceive motion [23,24,25,26,48,49,50].

Proprioception from the neck muscles have an important role in modulating the directional tuning patterns of deep cerebellar and vestibular neurons [51]. It is therefore possible that abnormal activity of the dystonic neck muscles, especially in those with robust dystonia, even without presence of tremor, putatively distorts proprioceptive output, introducing the noise in vestibular and proprioceptive convergence neurons in deep cerebellar and vestibular nuclei. Such an effect impacts the directional tuning pattern of deep cerebellar and vestibular neurons and would affect further upstream function of heading direction discrimination. According to this mechanism, the effect of dystonia and tremor is more robust and direct on vestibular heading direction discrimination, but less so in visual heading direction discrimination.

All the experiments, in visual and vestibular heading direction discrimination tasks, were performed while participants’ heads were immobilized in straight-ahead orientation. Such a setup excluded the possibility of asymmetrically stimulating the vestibular end-organs due to tonic head turning.

Although results provide important differences in vestibular and visual motion perception in patients with dystonia, tremor, and their combination; the results should be viewed as pilot. The sample size is small, given the relative rarity of isolated dystonia, isolated tremor, and their combination. Such small sample size limits our ability to perform rigorous statistics. Future studies may be needed to expand this sample size, looking at effects of jerky dystonia, tremor with dystonia, isolated tremor or head or hand (separately), and fixed dystonia on vestibular and visual heading perception. Future studies would be valuable to examine the effects of botulinum neurotoxin or deep brain stimulation on vestibular and visual heading perception in dystonia with or without tremor.

## 5. Conclusions

This report dually demonstrates shared deficits and distinctions in linear translational heading discrimination between individuals with isolated CD, CD with tremor, and isolated tremor. All three groups had reduced heading perception compared to healthy controls. The group with tremor (with or without CD) displayed less severe elevations in threshold detection. Collectively, the findings reveal overlapping navigation challenges, likely stemming from common cerebellar-brainstem dysfunctions, contrasted by nuances in impairment severity and characteristics across these movement disorders. This signifies fresh insights into altered spatial cognition with implications for individualized management, targeting core deficits. Additional inquiries across diverse patient cohorts with larger sample sizes and exploration of multi-faceted navigation will serve to enhance interventions that improve mobility and reduce associated fall risks in these cohorts.

## Figures and Tables

**Figure 1 brainsci-14-00217-f001:**
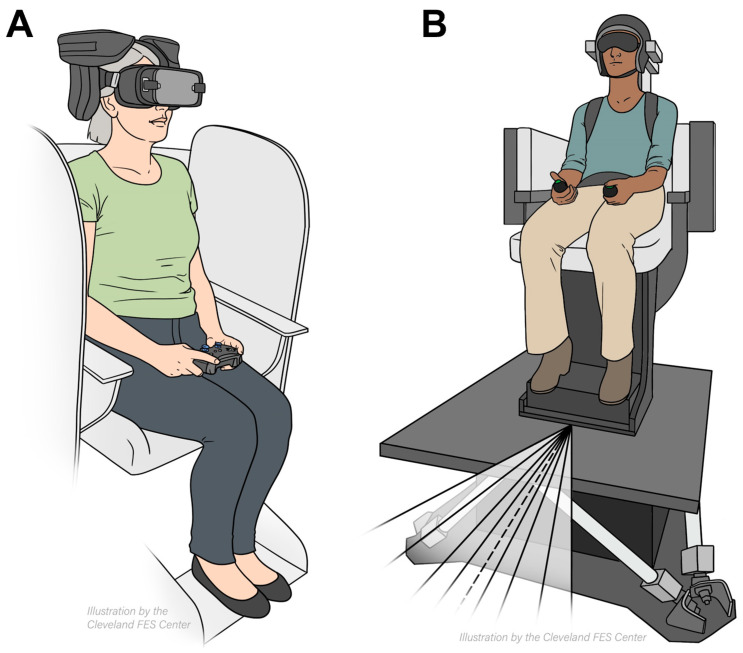
Schematic description of the experimental setup measuring visual (**A**) and vestibular (**B**) heading perception. The participant is seated on a chair mounted upon the motion platform and wears a VR headset. The vestibular-only experiments are performed within a dark room with the VR headset turned off and a pair of shades worn; visual-only experiments involve viewing of an animated conical star-cloud in virtual reality while the motion platform remains stationary.

**Figure 2 brainsci-14-00217-f002:**
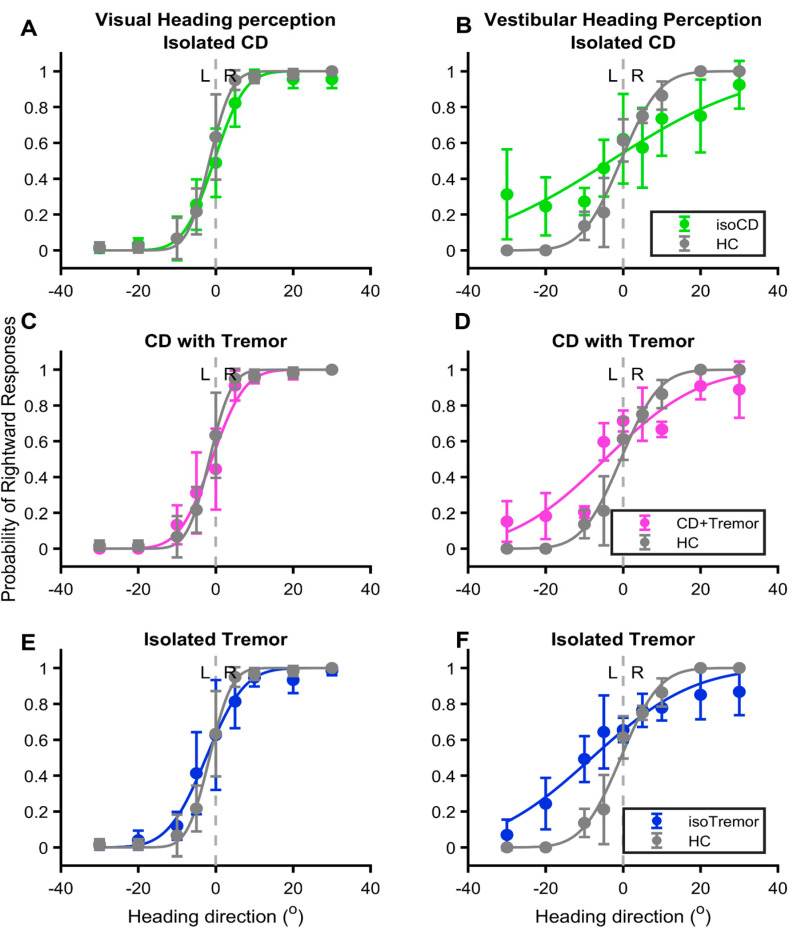
Examples of psychometric function curves objectively measuring visual (**A**,**C**,**E**) and vestibular (**B**,**D**,**F**) heading perception performance. Probability of rightward heading is depicted on the *y*-axis (1.0 highest and 0 lowest), while direction of heading is plotted on the *x*-axis (negative leftward, positive rightward). Isolated cervical dystonia patient’s responses are plotted in green dots, cervical dystonia with tremor is plotted with pink symbols, while isolated tremor is plotted with blue symbols. The healthy controls are gray symbols. Vertical dashed line depicts straight-ahead. Gray line depicts psychometric function fit for healthy controls.

**Figure 3 brainsci-14-00217-f003:**
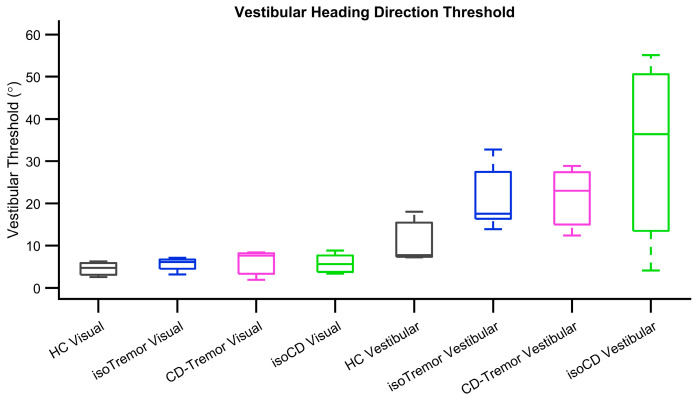
Summary of threshold values measured from isolated CD (green box-whisker plots), tremor with CD (pink box-whisker plots), isolated tremor (blue box-whisker plots) and healthy controls (gray box-whisker plots).

**Table 1 brainsci-14-00217-t001:** Clinical features and demographics.

Gender	Age	Duration	TWSTRS	Clinical Features of Dystonia
M	52	20	10	Mild right torticollis, Slight left laterocollis
M	43	11	23	Mild right torticollis, mild right laterocollis, right lateral shift
M	59	15	16	Slight right torticollis, left laterocollis, right lateral shift
M	62	10	21	mild right torticollis, mild right laterocollis, lateral shift
M	74	12	6	mild right torticollis, anterocollis
M	78	29	23	Mild right torticollis, anterocollis, mild sagittal shift
M	65	12	9	Right torticollis, with neck tremor.
M	74	15	10	Slight right torticollis, moderate to severe tremor
M	71	12	22	Slight anterocollis, neck tremor
M	72	20	NA	Moderate to severe tremor
M	31	3	NA	Bilateral hand tremor
M	50	8	NA	Bilateral hand and neck tremor
F	39	8	NA	Bilateral hand tremor
M	62	15	NA	Bilateral hand tremor

## Data Availability

Data will be available upon request after appropriate institutional agreement between requesting institution and United States Department of Veteran’s Affairs has been secured.

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
