# Peer review of "Exploring Heading Direction Perception in Cervical Dystonia, Tremor, and Their Coexistence"

_brainsci, 2024, doi:10.3390/brainsci14030217_

Round 1

Reviewer 1 Report

Comments and Suggestions for Authors

This is an interesting pilot study evaluating the role of vestibular function in cervical dystonia.  It is clear and well written.  Questions about study design:

1. There is no comment regarding the severity of dystonia in the patients when they were undergoing testing.  

2.  It is not clear if the patients were assessed in the beginning, middle or end of their botulinum toxin treatment cycle as this may also influence the outcomes of the study.

Author Response

Reviewer 1:

This is an interesting pilot study evaluating the role of vestibular function in cervical dystonia.  It is clear and well written.  Questions about study design:

  1. There is no comment regarding the severity of dystonia in the patients when they were undergoing testing.
  2. It is not clear if the patients were assessed in the beginning, middle or end of their botulinum toxin treatment cycle as this may also influence the outcomes of the study.

Response:

  1. Thank you for important feedback. Revised manuscript has a demographic table with details on severity of dystonia and tremor.
  2. The patients were assessed immediately before (or after) botulinum toxin injection – hence the effects of botulinum toxin was not present when data was collected.

Reviewer 2 Report

Comments and Suggestions for Authors

Thank you for the opportunity to review this manuscript, which presents a very interesting topic for neurology and neurootology. My considerations about the manuscript are described in detail below:

Abstract:

- I believe that adding subtopics (Background, objectives, methods, results and conclusion) will help better understand the text. I suggest including the subtopics in the abstract.

- The objective of the abstract is very vague, exploring is a very broad term that makes it difficult to understand what the authors were looking for in the study and the objective of the study seems not to have been answered in the abstract's conclusions. I suggest rewriting.

- The abstract is well written, however, the description of the methods was very brief, and did not provide details of which outcomes were evaluated. Therefore, when we read the results we cannot understand what was evaluated. I suggest improving the writing of the abstract methods by better describing what the authors evaluated in the volunteers.

- The results mention the findings, however, there is no mention of p-values. If there was a statistical analysis that demonstrated the p values, I suggest including it in each of the analyzed outcomes.

Introduction:

- The introduction is well written, with a logical and temporal sequence of the problem to be studied.

- The objective of the study in the abstract and at the end of the introduction is not the same. I suggest that authors standardize on a single study objective at the end of the introduction in the abstract.

Methods:

- What is the design/type of the study?

- How was the sample recruited? Was there a calculation to estimate the sample size, or is it a convenience sample? The authors need to further detail how the sample for this study was recruited.

- The description of the inclusion criteria for study volunteers is insufficient for readers to understand what the profile of these volunteers was. I suggest that the authors provide a greater and better description of the study inclusion criteria.

- When reading the entire method, one understands how data collection was carried out, however, some descriptions that should be in the data collection text were described in the statistical analysis. For example: I understood that the platform was moving and the volunteers needed to inform where this movement was using the control button, right? This needs to be in the description of data collection, not just in the statistical analysis, I suggest uploading the text to the description of data collection.

- In statistical analysis there is no description of how the data was analyzed. And this needs to be detailed. Were there any statistical tests used on the data? Are statistics just descriptive? This also needs to be better described and clarified in the manuscript text. I suggest including.

Results:

- As there was no description of the tests used in the statistical analysis, it was confusing and difficult to understand the results.

- The authors appear to compare groups, however, there is no mention or description of groups in the study methods.

- Furthermore, I missed a table that demonstrated the characterization of the sample, in relation to the volunteers' average age, height, weight, BMI and other variables, which may have been collected by the authors. I therefore suggest the inclusion of this table in the study.

Discussion:

- The discussion is very good and focused on what needs to be discussed, according to what the authors report. However, I missed making sense of what was found. Okay, the authors found these disorders, but what then? What do they suggest for these volunteers to improve these functional capabilities? Vestibular rehabilitation? Pharmacological treatment?

- Therefore, I suggest that the authors create a new paragraph providing directions to improve the volunteers' symptoms and ending with an indication of therapy, for the findings found in this study.

Conclusion:

- Authors should note that there is no mention of groups in the methods. Just a description of the patients' symptoms, if they were divided into groups according to the symptoms presented, this needs to be better described in the methods and statistical analysis of the data.

Author Response

Reviewer 2:

Abstract:

- I believe that adding subtopics (Background, objectives, methods, results and conclusion) will help better understand the text. I suggest including the subtopics in the abstract.

Response: Thank you for important suggestion. Subheadings are included as suggested.

- The objective of the abstract is very vague, exploring is a very broad term that makes it difficult to understand what the authors were looking for in the study and the objective of the study seems not to have been answered in the abstract's conclusions. I suggest rewriting.

Response: The abstract has been edited as suggested, emphasizing the objective of the study, and including more specific terms.

- The abstract is well written, however, the description of the methods was very brief, and did not provide details of which outcomes were evaluated. Therefore, when we read the results we cannot understand what was evaluated. I suggest improving the writing of the abstract methods by better describing what the authors evaluated in the volunteers.

Response: Thank you. The abstract has been rewritten to address the above and preceding comment.

- The results mention the findings, however, there is no mention of p-values. If there was a statistical analysis that demonstrated the p values, I suggest including it in each of the analyzed outcomes.

Response: We understand that. Given limited sample size we were not able to do accurate statistics. We have included this as a caveat in the revised manuscript.

Introduction:

- The introduction is well written, with a logical and temporal sequence of the problem to be studied.

- The objective of the study in the abstract and at the end of the introduction is not the same. I suggest that authors standardize on a single study objective at the end of the introduction in the abstract.

Response: Thank you. The comment is addressed as suggested.

Methods:

- What is the design/type of the study?

Response: This is an experimental study examining a physiological question in two disease classes; hence we hesitate to call is “case control” or “case series” as the pitch is (which related to disease class) not entirely clinical.

- How was the sample recruited? Was there a calculation to estimate the sample size, or is it a convenience sample? The authors need to further detail how the sample for this study was recruited.

Response: We recruited participants who visited our clinic, and were willing to enroll in study over last one and half year. Given limited numbers of patients (rarity of disease – dystonia is relatively rare, and pure tremor is rare to find as well) we could not select number of patients from the larger cohort.

- The description of the inclusion criteria for study volunteers is insufficient for readers to understand what the profile of these volunteers was. I suggest that the authors provide a greater and better description of the study inclusion criteria.

Response: We have emphasized the recruitment inclusion and exclusion criteria. Thank you.

- When reading the entire method, one understands how data collection was carried out, however, some descriptions that should be in the data collection text were described in the statistical analysis. For example: I understood that the platform was moving and the volunteers needed to inform where this movement was using the control button, right? This needs to be in the description of data collection, not just in the statistical analysis, I suggest uploading the text to the description of data collection.

Response: We have done best to re-organize the methods; but data collection/experimental setup has experiment events and data analysis has how collected data is analyzed. If we moved information from one place to the other it will make it confusing. Thank you.

- In statistical analysis there is no description of how the data was analyzed. And this needs to be detailed. Were there any statistical tests used on the data? Are statistics just descriptive? This also needs to be better described and clarified in the manuscript text. I suggest including.

Response: Thank you for this comment. CD is rare, and even more rare is CD with obvious tremor (sinusoidal oscillations). Tremor that is isolated without any neurological deficits is also rare (most tremor, although very common, and generally mixed). We were interested in CD, CD with tremor, and pure tremor. Hence we had limited dataset. Given such limitation in number, we have to be cautious about statistics application, hence we presented this data as descriptive.

Results:

- As there was no description of the tests used in the statistical analysis, it was confusing and difficult to understand the results.

Response: Thank you. Given the nature of dataset, the statistics had to be descriptive. Please see comment above.

- The authors appear to compare groups, however, there is no mention or description of groups in the study methods.

Response: The description of group is outlined in revised methods. They describe sutype – i.e. CD, CD with tremor; and CD without tremor.

- Furthermore, I missed a table that demonstrated the characterization of the sample, in relation to the volunteers' average age, height, weight, BMI and other variables, which may have been collected by the authors. I therefore suggest the inclusion of this table in the study.

Response: Thank you. We think most appropriate table would be clinical demographic table, it is now included in revised manuscript.

Discussion:

- The discussion is very good and focused on what needs to be discussed, according to what the authors report. However, I missed making sense of what was found. Okay, the authors found these disorders, but what then? What do they suggest for these volunteers to improve these functional capabilities? Vestibular rehabilitation? Pharmacological treatment?

- Therefore, I suggest that the authors create a new paragraph providing directions to improve the volunteers' symptoms and ending with an indication of therapy, for the findings found in this study.

Response: This is simply discussion of physiological differences in motion perception in two broad disease classes, and it indirectly suggests differences (and similarities) in pathophysiology of dystonia and tremor – but it is a extended implication of our findings. We kept the discussion appropriately conservative to address this issue.

Conclusion:

- Authors should note that there is no mention of groups in the methods. Just a description of the patients' symptoms, if they were divided into groups according to the symptoms presented, this needs to be better described in the methods and statistical analysis of the data.

Response: The groups were cervical dystonia versus isolated tremor and their combination. This is clarified further in the revised manuscript.

Reviewer 3 Report

Comments and Suggestions for Authors

1.     Please provide the type of study in the title. Also, it is suggested that the location of the study be included.

2.     Methods.

a.     Define the conditions; what are the patients with tremors? Where is the tremor?

b.     Patients with cervical dystonia have only cervical dystonia. Any other condition was not included, even a history of alcoholism and beta-agonist use?

c.     From where came the 19 individuals? All from the same institution?

d.     Regarding the experiment, were the individuals previously advised of what would appear? How were they advised regarding the procedure?

3.     Statistics

a.     Please describe the distribution. In the description, the authors used a Gaussian distribution.

b.     Was it used in the software for the analysis?

c.     How was the power of the study performed? Why was this specific number of individuals with Cd, CD + tremor, unspecific tremor, and healthy?

4.     Results

a.     Please provide baseline characteristics of the individuals in the study.

5.     Please answer all the queries after the conclusion, including supplementary material, author contribution, funding, IRB, and consent statement.

6.     Also, include the IRB number at the end and in the methodological description.

7.     See if reference 53 is pertinent. It is probably a typo.

Author Response

Reviewer 3:

  1. Please provide the type of study in the title. Also, it is suggested that the location of the study be included.

Response: Thank you for your suggestion. Given the nature of this (physiological/experimental) study, we do not feel appropriate to put type of study and location of the study in the title – it does not add any information that is important for “screeners” or “readers” to know. But happy to get more input if needed.

  1. Methods.
  2. Define the conditions; what are the patients with tremors? Where is the tremor? Response: Tremor and CD are defined in first paragraph of introduction.
  3. Patients with cervical dystonia have only cervical dystonia. Any other condition was not included, even a history of alcoholism and beta-agonist use?

Response: Yes these were isolated CD. No alcoholism (that was chronic and pathological) or beta-agonist use for these patients.

  1. From where came the 19 individuals? All from the same institution?

Response: All came from movement disorders clinic at Cleveland VA Medical Center.

  1. Regarding the experiment, were the individuals previously advised of what would appear? How were they advised regarding the procedure?

Response: Yes, they were all instructed and given prior instructions of what is experiment comprised of.

  1. Statistics
  2. Please describe the distribution. In the description, the authors used a Gaussian distribution.
  3. Was it used in the software for the analysis?
  4. How was the power of the study performed? Why was this specific number of individuals with Cd, CD + tremor, unspecific tremor, and healthy?

            Response: Participants were selected from our clinic, those who had met the criteria and were willing to participate were offered to join the study. Given limited numbers of patients (rarity of disease – dystonia is relatively rare, and pure tremor is rare to find as well) we could not select number of patients from the larger cohort. Given sample size, we were able to perform descriptive analysis only.

  1. Results
  2. Please provide baseline characteristics of the individuals in the study.
  3. Please answer all the queries after the conclusion, including supplementary material, author contribution, funding, IRB, and consent statement.
  4. Also, include the IRB number at the end and in the methodological description.
  5. See if reference 53 is pertinent. It is probably a typo.

Response: Thank you for all the suggestions in point 4-7 above. We have addressed them all in revised manuscript.

Reviewer 4 Report

Comments and Suggestions for Authors

The authors reported an interesting study on patients with Cervical Dystonia and Tremor. I have some comments to the authors:

- Please specify in the introduction that cervical dystonia is the most common form of dystonia in adult. The authors should expand the description of dystonia, including the definition of sensory trick, the possibility that dystonia may spread to another body site etc.

- The authors should also refer to the recent clinical diagnostic guideline in CD:

Defazio G, et al. Validation of a guideline to reduce variability in diagnosing cervical dystonia. J Neurol. 2023 

- Please specify how you perform the diagnosis of essential tremor in your patients. In this regard the authors should also state that none of the patients suffered from functional dystonia or functional tremor. 

- The authors should provide a table with the clinical and demographic features of the patients, including disease duration, education and gender.

- The authors should add a paragraph with limitations of the study.

Author Response

Reviewer 4:

- Please specify in the introduction that cervical dystonia is the most common form of dystonia in adult. The authors should expand the description of dystonia, including the definition of sensory trick, the possibility that dystonia may spread to another body site etc.

Response: Thank you. Dystonia is defined as suggested. Brief phenomenology is also described in the revised version.

- The authors should also refer to the recent clinical diagnostic guideline in CD:

Defazio G, et al. Validation of a guideline to reduce variability in diagnosing cervical dystonia. J Neurol. 2023 

Response: Thank you. Reference cited as suggested.

- Please specify how you perform the diagnosis of essential tremor in your patients. In this regard the authors should also state that none of the patients suffered from functional dystonia or functional tremor. 

Response: Thank you. Definition of tremor and dystonia and inclusion criteria outlines this fact, and it is included in revised manuscript.

- The authors should provide a table with the clinical and demographic features of the patients, including disease duration, education and gender.

Response: Table is provided as suggested. Thank you.

- The authors should add a paragraph with limitations of the study.

Response: Thank you. Key limitations and future directions are outlined in a dedicated paragraph.

Round 2

Reviewer 2 Report

Comments and Suggestions for Authors

The authors did a good job, all my questions were answered by the authors and modified in the manuscript. Therefore, I believe that this latest version of the manuscript has a clearer and more concise text, contributing to greater and better interpretation by readers and is ready to be accepted for publication.

Reviewer 3 Report

Comments and Suggestions for Authors

  The corrections are satisfactory.

Reviewer 4 Report

Comments and Suggestions for Authors

The authors have address all the points.